# Oligosaccharide Metabolism and Lipoteichoic Acid Production in *Lactobacillus gasseri* and *Lactobacillus paragasseri*

**DOI:** 10.3390/microorganisms9081590

**Published:** 2021-07-26

**Authors:** Tsukasa Shiraishi, Shintaro Maeno, Sayoko Kishi, Tadashi Fujii, Hiroki Tanno, Katsuaki Hirano, Takumi Tochio, Yasuhiro Tanizawa, Masanori Arita, Shin-ichi Yokota, Akihito Endo

**Affiliations:** 1Department of Microbiology, Sapporo Medical University School of Medicine, Sapporo 060-8556, Hokkaido, Japan; syokota@sapmed.ac.jp; 2Department of Food, Aroma and Cosmetic Chemistry, Faculty of Bioindustry, Tokyo University of Agriculture, Abashiri 099-2493, Hokkaido, Japan; maeno.shintaro.235@m.kyushu-u.ac.jp (S.M.); kishi.biken.0801@gmail.com (S.K.); 47620003@nodai.ac.jp (H.T.); 3B Food Science Co., Ltd., Chita 478-0046, Aichi, Japan; t-fujii@bfsci.co.jp (T.F.); k-hirano@bfsci.co.jp (K.H.); t-tochio@bfsci.co.jp (T.T.); 4Center for Information Biology, National Institute of Genetics, Mishima 411-8540, Shizuoka, Japan; ytanizaw@nig.ac.jp (Y.T.); arita@nig.ac.jp (M.A.); 5RIKEN Center for Sustainable Resource Science, Yokohama 230-0045, Kanagawa, Japan

**Keywords:** *Lactobacillus gasseri*, *Lactobacillus paragasseri*, kestose, lipoteichoic acid, glycoside hydrolase, RNAseq

## Abstract

*Lactobacillus gasseri* and *Lactobacillus paragasseri* are human commensal lactobacilli that are candidates for probiotic application. Knowledge of their oligosaccharide metabolic properties is valuable for synbiotic application. The present study characterized oligosaccharide metabolic systems and their impact on lipoteichoic acid (LTA) production in the two organisms, i.e., *L. gasseri* JCM 1131^T^ and *L. paragasseri* JCM 11657. The two strains grew well in medium with glucose but poorly in medium with raffinose, and growth rates in medium with kestose differed between the strains. Oligosaccharide metabolism markedly influenced their LTA production, and apparent molecular size of LTA in electrophoresis recovered from cells cultured with glucose and kestose differed from that from cells cultured with raffinose in the strains. On the other hand, more than 15-fold more LTA was observed in the *L. gasseri* cells cultured with raffinose when compared with glucose or kestose after incubation for 15 h. Transcriptome analysis identified glycoside hydrolase family 32 enzyme as a potential kestose hydrolysis enzyme in the two strains. Transcriptomic levels of multiple genes in the *dlt* operon, involved in D-alanine substitution of LTA, were lower in cells cultured with raffinose than in those cultured with kestose or glucose. This suggested that the different sizes of LTA observed among the carbohydrates tested were partly due to different levels of alanylation of LTA. The present study indicates that available oligosaccharide has the impact on the LTA production of the industrially important lactobacilli, which might influence their probiotic properties.

## 1. Introduction

Prebiotics are defined as a substrate that is selectively utilized by host microorganisms conferring a health benefit [1]. Oligosaccharides are the best characterized prebiotics. Administration of prebiotics promotes the growth of beneficial bacteria in the complex gut microbiota, resulting in beneficial effects in the host through host–microbe interactions and microbial metabolites [1]. Short-chain fructooligosaccharides (FOSs) are well-studied and commercialized prebiotics. They are usually commercialized as a mixture of degrees of polymerization (DP) of 3 and 4—i.e., 1-kestose [kestose, Fur (β2–1) Fur (β2–1α) Glc] and nystose [Fur (β2–1) Fur (β2–1) Fur (β2–1α) Glc]—respectively, with trace DP5 FOS, fructosylnystose. Among the components, kestose is a promising prebiotic for the stimulation of beneficial gut microbes, including bifidobacteria and butyrate producers [2,3]. Although the impact of oligosaccharides on the growth of beneficial gut microbes and probiotic strains is widely known, limited studies have addressed the impact of oligosaccharides, especially kestose, on the physiological properties of such bacteria. Different carbohydrates activate the expression of different metabolism-related and -unrelated proteins in microbes [4,5,6], and such proteins are often essential for host-microbe interactions.

*Lactobacillus gasseri* is a human commensal lactic acid bacterium and certain strains of the species are well characterized as probiotics. Several beneficial outcomes have been reported after administration of the strains [7]. Recent taxonomic studies revealed that *L. gasseri* strains are genomically divided into two groups, and one group was reclassified as *Lactobacillus paragasseri* sp. nov. [8]. This suggests that the accumulated knowledge of probiotic properties of *L. gasseri* might be applicable to *L. paragasseri*. Indeed, over 85% of 92 *L. gasseri*-like strains isolated from human feces in China were classified as *L. paragasseri* [9]. *Lactobacillus paragasseri* strain K7 suppressed the NF-κB pathway via TLR10-mediated signaling, in addition to the activation of the TLR2-mediated phosphoinositide-3-kinase/Akt pathway and protein kinase C pathway, resulting in reduced necrosis/apoptosis and increased intestinal epithelial barrier integrity, respectively [10]. *Lactobacillus gasseri* actively grows in the presence of kestose but poorly with nystose [11], but such metabolic properties have not been studied for *L. paragasseri*. FOS metabolic systems in the organisms have not been characterized even though they are important for the application of synbiotics.

*Lactobacillus gasseri* cells contain lipoteichoic acid (LTA). LTA is a cell surface molecule of Gram-positive bacteria and is anchored to the cell membrane by a glycolipid moiety. LTA is typically composed of poly-glycerophosphate (GroP) linked to glycolipids through phosphodiester linkages. The hydroxyl groups of GroP repeating units can be partially substituted by D-alanine, D-glucose, D-galactose, and/or *N*-acetyl-D-glucosamine. LTA in *L. gasseri* JCM 1131^T^ has a unique glycolipid anchor structure with a tetrasaccharide as the carbohydrate moiety [12]. LTA found in probiotics has been linked to several beneficial activities of the microbes, including immune reaction [13]. D-Alanine residues of the GroP repeating units and fatty acid residues of the glycolipid moiety in the LTA chemical structure are essential for immune reaction [14,15]. Growth conditions affect the chemical structure of LTA [16,17,18], which may lead to the different probiotic properties of lactobacilli.

In the present study, LTA production by *L. gasseri* JCM 1131^T^ and *L. paragasseri* JCM 11657 was assessed in the presence of different carbohydrates—i.e., glucose, kestose, and raffinose [Gal (α1–6) Glc (α1–2β) Fur]—to study the impact of available oligosaccharide on physiological properties of industrially important lactobacilli. Carbohydrate metabolic systems were also characterized to consider efficient prebiotic application in these strains.

## 2. Materials and Methods

### 2.1. Bacterial Strains and Culture Conditions

*Lactobacillus gasseri* JCM 1131^T^ and *L. paragaseri* JCM 11657 were obtained from the Japan Collection of Microorganisms, RIKEN BioResource Center (Ibaraki, Japan). The two strains were routinely pre-cultured in Difco Lactobacilli MRS Broth (Becton, Dickinson and Co., Franklin Lakes, NJ, USA) supplemented with 0.05% (*w*/*v*) L-cysteine-HCl at 37 °C overnight under anaerobic conditions using AnaeroPack-Anaero (Mitsubishi Gas Chemical Company, Inc., Tokyo, Japan), and used in the following studies. 

### 2.2. Oligosaccharide Metabolic Properties

The pre-cultured cells were washed three times with saline and resuspended in saline, and the bacterial cell suspension was prepared in saline at an optical density (OD) value at 660 nm of 1.0 with a spectrophotometer (Model U-2001, (Hitachi, Ltd., Tokyo, Japan)). The cell suspension was inoculated into tested broth at a ratio of 0.1% (*v*/*v*) and incubated for 15 h at 37 °C under the anaerobic conditions. The tested broth was composed of (*l*^−1^) 2.8 g of Difco Thioglycollate Medium Without Dextrose or Indicator (Becton, Dickinson and Co.), 1.0 g of Oxoid Lab-Lemco Powder (Thermo Fisher Scientific Inc., Waltham, MA, USA), 0.2 g of ammonium citrate, 1.0 g of sodium acetate, 0.05 g of Tween 80, 0.04 g of MgSO_4_–7H_2_O, 0.002 g of MnSO_4_–4H_2_O, 0.002 g of FeSO_4_–7H_2_O, 0.002 g of NaCl, and 2.0 g of carbohydrate (pH 5.8). Carbohydrates included were glucose (98% purity, Wako Pure Chemical Industries, Ltd., Osaka, Japan), kestose (99% purity, B Food Science Co., Ltd., Aichi, Japan), and raffinose (98% purity, Wako Pure Chemical Industries, Ltd.), and sugar-free medium served as a negative control. Raffinose, a common prebiotic, was used for a reference. Growth was monitored using the OD value at 660 nm with a spectrophotometer.

### 2.3. Immunoblotting

Sodium dodecyl sulfate-polyacrylamide gel electrophoresis (SDS-PAGE) and immunoblotting were performed to detect LTA in cultured cells, as described elsewhere [19]. Briefly, bacterial cell suspensions were treated at 100 °C for 5 min in SDS-PAGE sample buffer. Cell suspensions were further incubated with 1 mg/mL of proteinase K (Wako Pure Chemical Industries, Ltd.) for 60 min at 60 °C. The incubated suspensions were applied to electrophoresis for 45 min at 30 mA on a 15% (*w*/*v*) polyacrylamide gel, and then transferred to a polyvinylidene difluoride (PVDF) membrane (EMD Millipore Corp., Billerica, MA) by electroblotting for 60 min at 100 V. Prestained XL-Ladder Broad (APRO Life Science Institute, Inc., Tokushima, Japan) was used as a reference for the relative molecular size. The blotted membrane was blocked with 5% (*w*/*v*) skim milk in phosphate-buffered saline (PBS) containing 0.05% (*v*/*v*) polysorbate 20 for 60 min at room temperature. The membrane was incubated with a 1:500 dilution of murine anti-LTA monoclonal antibody (mAb) clone 55 (Hycult Biotech, Uden, The Netherlands) for 90 min at room temperature, and subsequently reacted with a 1:1000 dilution of alkaline phosphatase-conjugated goat anti-mouse immunoglobulin antibody (BioSource International, Inc., Camarillo, CA) for 90 min at room temperature. Specific binding was visualized with 0.1 M sodium carbonate buffer (pH 9.6) containing 0.15 mg/mL of 5-bromo-4-chloro-3-indolyl phosphate *p*-toluidine salt, 0.3 mg/mL of nitroblue tetrazolium, and 1 mM MgCl_2_.

### 2.4. Quantification of LTA

LTA was purified from *L. gasseri* JCM 1131^T^ by 1-butanol extraction followed by hydrophobic interaction chromatography, as described previously [19], and was used to develop a calibration curve. Immunoblotted membrane images were captured and exported to ImageJ software (Java-based image processing program, National Institutes of Health (NIH), Bethesda, MD, USA and the Laboratory for Optical and Computational Instrumentation (LOCI), University of Wisconsin, Madison, WI, USA) to quantify LTA in the bacterial cell suspensions using the calibration curve. One-way ANOVA and Tukey’s post hoc test were applied to compare the amount of LTA among the cells cultured with different carbohydrates using Statcel software (OMS Publishing, Saitama, Japan). A *p*-value of 0.05 was considered significant.

### 2.5. Draft Genome Sequencing and Acquisition of Genomic Data

Whole-genome sequencing of *L. paragasseri* JCM 11657 was conducted by Illumina NovaSeq 6000 (Illumina, Inc., San Diego, CA). Reads were assembled using Platanus_B (version 1.1.0) with default settings [20]. Sequences shorter than 300 bp were eliminated. The genome was annotated using the DDBJ Fast Annotation and Submission Tool (DFAST, https://dfast.nig.ac.jp, accessed on 14 February 2019) [21]. The complete genome sequence of ATCC 33323^T^ (=JCM 1131^T^) was obtained from DFAST Archive of Genome Annotation (DAGA, https://dfast.nig.ac.jp, accessed on 26 January 2021) [21].

### 2.6. Genome Analysis

Genome level identities of the two strains and reference strains were determined by calculating the average nucleotide identity (ANI), as described previously [22]. The genomic data of the strains were used to search for glycoside hydrolase (GH) family enzymes using dbCAN2 in the CAZy database with HMMER, DIAMOND, and Hotpep tools. GH proteins were identified when detected by two of the three tools, as recommended by the database [23].

### 2.7. RNA Sequencing of L. gasseri and L. paragasseri

RNA sequencing was conducted to clarify responses to different carbohydrates in *L. gasseri* JCM 1131^T^ and *L. paragasseri* JCM 11657. The pre-cultured cells were washed three times with saline and diluted in saline to an OD value at 660 nm of 1.0. Two-hundred microliter of the diluted cells was inoculated into 1.8 mL of tested broth and cultured for 6 h at 37 °C under the anaerobic conditions. The tested broth was the broth used to study the oligosaccharide metabolic properties, and glucose, kestose, and raffinose were included. RNA was extracted from the 6-h-old cells (mid- to late-logarithmic phase) of three different cultures using the combination of RNAprotect Bacteria Reagent and RNeasy Mini Kit (Qiagen, Venlo, The Netherlands) according to the manufacturer’s instructions. The isolated RNA was sent to Macrogen Japan Corp. (Tokyo, Japan) and used for sequencing. Present rRNA in the samples was removed by the Ribo-Zero rRNA Removal Kit designed for bacteria (Illumina), and the resulting product was used to prepare mRNA libraries using TruSeq Stranded mRNA (Illumina) according to the manufacturer’s instructions. The libraries were sequenced using Illumina NovaSeq 6000 with 150-bp paired-end sequencing. The quality of sequences was verified using FastQC (version 3, https://www.bioinformatics.babraham.ac.uk/projects/fastqc/, accessed on 4 February 2021). Trimmomatic (version 0.36) [24] was used to remove adapter sequences. The obtained clean sequences were mapped to the genomes of the two strains using Bowtie2 (version 2.3.5.1) [25], and assembled and normalized based on the transcripts per million (TPM) approach [26] using StringTie (version 2.0.4) [27]. Statistical differential expression tests were performed with DESeq2 (version 1.28.1) [28]. The genes were considered to be differentially expressed when their adjusted *p*-values and absolute log_2_ fold changes (FCs) were < 0.05 and > 1, respectively. The genes differentially expressed were assigned to Cluster of Orthologous Groups (COG) functional classification using the COGNITOR software [29]. Transcriptomic data of the strains in the presence of different carbohydrates were included to conduct principal component analysis (PCA) and Ward.D2 hierarchical clustering using the TCC-GUI [30].

## 3. Results

### 3.1. Growth in the Presence of Different Carbohydrates

*Lactobacillus gasseri* JCM 1131^T^ and *L. paragasseri* JCM 11657 were cultured in the presence of glucose, kestose, or raffinose. *Lactobacillus gasseri* grew at a similar level when glucose or kestose was present. Little growth was recorded in the presence of raffinose (Figure 1). For *L. paragasseri*, the best growth was recorded in the presence of glucose and growth in the presence of kestose was also observed. However, growth was poor when raffinose was supplemented (Figure 1).

### 3.2. Detection of LTA in Cells Cultured with Different Carbohydrates

The apparent molecular size of LTA on SDS-PAGE differed among cells cultured with different carbohydrates. For both *L. gasseri* JCM 1131^T^ and *L. paragasseri* JCM 11657 strains, the apparent molecular size of LTA was similar between cells cultured with glucose and kestose, but the size larger than that from cells cultured with raffinose (Figure 2a).

The amount of LTA showed different tendencies between the strains. In *L. gasseri*, similar levels of LTA were produced in cells cultured with glucose or kestose, whereas more than 15-fold more LTA was produced in cells cultured with raffinose (*p* < 0.01) (Figure 2b). No significant differences in LTA production were observed in *L. paragasseri* JCM 11657 cells cultured with different carbohydrates (Figure 2c).

### 3.3. Genomic Characteristics

*Lactobacillus gasseri* JCM 1131^T^ (=ATCC 33323^T^) and *L. paragasseri* JCM 11657 had 1.89 Mbp and 1.93 Mbp genomes containing 1808 and 1852 coding sequences (CDSs), respectively. *Lactobacillus gasseri* JCM 1131^T^ had ANI values of 93.8% to *L. paragasseri* JCM 5343^T^, whereas *L. paragasseri* JCM 11657 had ANI values of 98.7% to *L. paragasseri* JCM 5343^T^ and 93.7% to *L. gasseri* ATCC 33323^T^. Thus, the tested strains were properly identified.

#### 3.3.1. Genes Involved in Kestose Metabolism

CAZy analysis revealed that *L. gasseri* JCM 1131^T^ and *L. paragasseri* JCM 11657 possess 30 and 31 GH family enzymes, respectively (Appendix A). Of the GH family enzymes, two GH32 enzymes and a single GH68 enzyme were found in *L. gasseri* as putative kestose degradation enzymes, and a single GH32 and GH68 enzyme were found in *L. paragasseri*. GH32 enzymes in the strains were sucrose-6-phosphate hydrolase (S6PH) and genes encoding phosphotransferase systems (PTSs) were adjacent to those encoding the S6PHs. S6PH is one of the most characterized enzymes to hydrolyze sucrose and kestose with or without phosphorylation in gut microbes [3]. Alignment of amino acid sequences of the GH32 proteins revealed that one of the GH32 proteins in *L. gasseri* (gene ID, 397) had one amino acid substitution (N to G) in the NDPNG motif (GDPNG) (Figure 3), which is one of the key regions for its hydrolase activity of GH32 [31], whereas others conserved this motif. GH68 enzymes, annotated as levansucrase, in the two strains contained a signal peptide, meaning that they are extracellular GH enzymes, whereas genes encoding the enzymes were divided by the presence of stop codon (GCA_000014425.1_01268 and GCA_000014425.1_01269). This is consistent with a previous report on GH68 of *L. gasseri* DSM 20243^T^ (=ATCC 33323^T^ =JCM 1131^T^) [32]. Complete GH68 (without a stop codon) has been reported in two other *L. gasseri* strains [32]. One part of the divided GH68 (GCA_000014425.1_01269) in *L. gasseri* JCM 1131^T^ comprised 651 amino acids containing a signal sequence and catalytic domain, whereas another part (GCA_000014425.1_01268) was 123 amino acid residues containing a cell-wall anchoring motif. Similarly, one part of the divided GH68 (locus_tag, PGA11657_05800) in *L. paragasseri* JCM 11657 comprised 646 amino acids containing a signal sequence and catalytic domain, whereas another part (PGA11657_05790) was 171 amino acid residues containing a cell-wall anchoring motif.

#### 3.3.2. Genes Involved in LTA Synthesis

Two well-characterized gene operons involved in the biosynthesis of LTA were found in the genomes of the two strains. The LTA biosynthetic operon [33] contained four genes encoding phosphoglycerol transferase (LTA synthase, LtaS), membrane protein, and two glycosyl transferase (GT) family 4 enzymes. The *dlt* operon, which is involved in the biosynthesis of D-alanine substitution of LTA [34] contained five genes—i.e., *dltA*, *dltB*, *dltC*, *dltD*, and *dltX*. Organization of the two operons was the same in *L. gasseri* JCM 1131^T^ and *L. paragasseri* JCM 11657. On the other hand, glycosyltransferase UgtP, which is involved in determining the molecular size of LTA in *Staphylococcus aureus* [35], was not observed in the two strains.

### 3.4. Transcriptomic Analysis

Transcriptomic analysis was conducted for the two strains cultured with glucose, kestose, or raffinose. Hierarchical clustering based on the transcriptomic data revealed that the transcriptome of *L. gasseri* JCM 1131^T^ cells in the presence of glucose or kestose was similar but distinct from that in the presence of raffinose (Figure 4a). *Lactobacillus paragasseri* JCM 11657 exhibited a similar tendency, but its transcriptome in the presence of glucose or kestose had lower identities than that of *L. gasseri* JCM 1131^T^ (Figure 4b). Principal component analysis (PCA) was consistent with the clustering (Figure 4c,d).

#### 3.4.1. Transcriptomic Analysis of Genes Involved in Carbohydrate Metabolism

In *L. gasseri* JCM 1131^T^, the number of differentially expressed genes—i.e., sum of number of upregulated genes—was only 69 between glucose and kestose, but it was 1,065 between glucose and raffinose, and 1,084 between kestose and raffinose (Table 1), accounting for 58% and 60% of all CDSs, respectively. In *L. paragasseri* JCM 11657, the number of differentially expressed genes was 468, 727, and 629 between glucose and kestose, between glucose and raffinose, and between kestose and raffinose, respectively (Table 1). The differentially expressed genes were associated with 21 COG functional categories. Of the 69 differentially expressed genes between glucose and kestose in *L. gasseri*, 13 (accounting for 19% of the differentially expressed genes) were assigned to class G for carbohydrate transport and metabolism (Table 2). Class G again accounted for the largest proportion (approx. 10%) of the differentially expressed genes between glucose and kestose in *L. paragasseri*. Class J for translation, ribosomal structure, and biogenesis accounted for the largest proportion between glucose and raffinose and between kestose and raffinose in both strains, except that class G for carbohydrate transport and metabolism accounted for the largest proportion between kestose and raffinose in *L. paragasseri*.

The divided GH68 (levansucrase) gene was the most upregulated gene in *L. gasseri* cells cultured with kestose when compared with those cultured with glucose, and absolute log_2_ FC values were over 4 (Table 3). The transcription of GH68 gene was higher in the order of kestose, raffinose, and glucose. In the two genes encoding S6PHs (GH32) of *L. gasseri*, one (gene ID, 1784) with the conserved NDPNG motif was upregulated in the presence of kestose when compared with glucose or raffinose, but the other (gene ID, 397) with one amino acid substitution in the motif (GDPNG) was not. The gene ID 1784 was also upregulated in the presence of raffinose compared with glucose. Genes encoding phosphotransferase systems (PTSs) adjacent to the genes encoding the two S6PHs demonstrated similar statistical profiles to those of the two S6PHs.

In *L. paragasseri* JCM 11657, the divided GH68 (levansucrase) gene containing a stop codon was the most upregulated gene in the presence of kestose when compared with glucose and the absolute log_2_ FC was over 4 (Table 4), similar to *L. gasseri* JCM 1131^T^. In addition, transcription of these genes was higher in the order of culture with kestose, raffinose, and glucose. A single GH32 (S6PH) gene in *L. paragasseri* was induced in the presence of kestose. On the other hand, sucrose phosphorylase (GH13) and its adjacent PTS, which were not observed in JCM 1131^T^, were strongly upregulated in the presence of raffinose in JCM 11657. The absolute log_2_ FCs of the two genes between culture with raffinose and glucose were 9.8 and 8.6, respectively, and these genes were the most upregulated in the presence of raffinose when compared with glucose or kestose. A large gene cluster (gene ID ranging from 14610 to 14690) containing genes encoding three GH13 enzymes, a single GH65 enzyme, and sugar ABC transporters was highly upregulated in the presence of glucose when compared with kestose.

#### 3.4.2. Transcriptomic Analysis of Genes Involved in LTA Synthesis

The transcriptome of two gene operons involved in the biosynthesis of LTA—i.e., LTA biosynthetic operon and *dlt* operon—was characterized in the two strains. In *L. gasseri*, three of the five genes in the *dlt* operon—i.e., *dltX*, *dltA*, and *dltB*—were upregulated in the presence of glucose when compared with raffinose, and *dltX* and *dltA* were upregulated in the presence of kestose when compared with raffinose (Table 3). The five genes were not differentially expressed when compared between culture with glucose and kestose. In *L. gasseri*, two genes encoding GT4 enzymes in the LTA biosynthetic operon were upregulated in the presence of glucose or kestose when compared with raffinose. In *L. paragasseri*, all five genes and three (*dltB*, *dltC*, and *dltD*) of the five genes in the *dlt* operon were upregulated in the presence of glucose when compared with raffinose and kestose, respectively (Table 4). Three (*dltX*, *dltA*, and *dltB*) of the five genes in the *dlt* operon were upregulated in the presence of kestose when compared with raffinose. Genes in the LTA biosynthetic operon of *L. paragasseri* were not differentially expressed in the presence of different carbohydrates.

## 4. Discussion

The available carbohydrates have a significant impact on the growth of probiotic bacteria, but little is known about the impact on the physiological properties of probiotics. Moreover, carbohydrate metabolic systems have only been partially studied in these organisms. The present study characterized oligosaccharide metabolic systems and the impact of oligosaccharides on LTA production in *L. gasseri* and *L. paragasseri*.

*Lactobacillus gasseri* JCM 1131^T^ and *L. paragasseri* JCM 11657 metabolized kestose but not raffinose. The two strains possessed one or two genes encoding S6PH classified as GH32 for the potential hydrolysis of kestose. *Lactobacillus gasseri* contained two S6PHs. One with gene ID 1784 conserved the NDPNG motif, but the other with gene ID 397 had one amino acid substitution (GDPNG) (Figure 3). A previous study reported that amino acid replacement of the initial N to S in this region significantly reduced the activity of the bacterial GH32 enzyme [36]. Genes encoding the NDPNG-conserved S6PH (gene ID 1784) and its adjacent PTS (gene ID 1783) were highly transcribed in the presence of kestose when compared with glucose (Table 3), but another S6PH (gene ID 397) and adjacent PTS (gene ID 396) were not. A single S6PH and its adjacent PTS in *L. paragasseri* were also significantly transcribed in the presence of kestose when compared with glucose (Table 4). This suggests that the NDPNG-conserved S6PH (gene ID 1784) and its adjacent PTS (gene ID 1783) are essential for the transportation and hydrolysis of kestose in *L. gasseri* and *L. paragasseri*. Of note, genes encoding GH68 levansucrase, which were present in a truncated form lacking the cell anchoring domain due to a stop codon in both strains, were the most highly induced in the presence of kestose when compared with glucose. Intact GH68 in other *L. gasseri* strains exhibits transglycosylation activity in the synthesis of fructan rather than hydrolysis activity, but such activity was absent in *L. gasseri* DSM 20243^T^ (=JCM 1131^T^) [32]. It is unknown whether the truncated form of GH68 protein is translated and has enzymatic activity.

Raffinose was not metabolized by the strains, whereas several genes encoding GH enzymes were significantly induced by the presence of raffinose when compared with glucose or kestose (Table 3 and Table 4). In particular, genes encoding GH13 sucrose phosphorylase and its adjacent PTS in *L. paragasseri* JCM 11657 were the most highly induced genes in the presence of raffinose when compared with glucose and kestose. GH13 sucrose phosphorylase in combination with GH36 α-galactosidase have been reported as the key enzymes for the degradation of raffinose [37], but the GH36 α-galactosidase was not induced by raffinose in *L. gasseri* or *L. paragasseri* (Table 4). This suggests that the GH36 α-galactosidases in the strains have different substrates, and this may be a reason for the deficiency of raffinose metabolic properties in the strain. A large gene cluster (gene IDs ranging from 14610 to 14690), containing genes encoding three GH13 enzymes and GH65, was highly transcribed in the presence of glucose and raffinose when compared with kestose, and transcription levels of most of these genes were similar between cells cultured with glucose and raffinose. The gene cluster was characterized by the degradation of starch and maltooligosaccharides in a phylogenetically related taxon, *Lactobacillus acidophilus* [38], and was not used for the metabolism of glucose or raffinose. This suggests that the different transcription levels observed in the present study were due to repression of these genes by kestose.

LTA production in the tested strains was strongly influenced by the carbohydrates available. In *L. gasseri*, more than 15-fold more LTA was produced in cells cultured with raffinose when compared with those cultured with glucose or kestose (Figure 2b). Moreover, differences in apparent molecular size on SDS-PAGE were observed in *L. gasseri* and *L. paragasseri* between cells cultured with glucose/kestose and raffinose (Figure 2a). Available carbohydrates, pH, and size of inoculum influence the amount of LTA in cells of several lactobacilli and streptococci [39,40,41]. LTA production increases in a strain/species-dependent manner [41]. The different sizes of LTA between culture conditions have been poorly reported in lactobacilli, but they may be partially characterized by the substitution rate of D-alanine residues bound to glycerol-phosphate polymers of LTA, as described for *S. aureus*. D-Alanylation of LTA is influenced by several environmental factors—including pH, temperature, and NaCl concentration—in *S. aureus* [42,43,44]. The different carbohydrates led to different growth rates and pH due to the accumulation of lactate in the tested strains (data not shown). These may be factors affecting the LTA production levels and molecular size of LTA in the tested strains. Our hypothesis was generally supported by the transcriptome analysis. Multiple genes in the *dlt* operon involved in D-alanylation of LTA were significantly transcribed in cells cultured with glucose/kestose when compared with raffinose. Different transcriptomic levels of the *dlt* operon and reduced D-alanylation of LTA have been reported in *S. aureus* [44]. Moreover, the levels of gene expression of the LTA biosynthetic operon containing GroP transferase (LtaS) in *L. paragasseri* were similar among the carbohydrates tested, being consistent with the levels of LTA production. On the other hand, the amount of LTA significantly increased in *L. gasseri* cells cultured with raffinose, but two glycosyltransferases (GT4) presumably involved in the synthesis of LTA anchor glycolipid were significantly transcribed in cells cultured with glucose/kestose when compared with raffinose. Transcription levels of LtaS involved in the synthesis of GroP polymer was similar between cells cultured with glucose/kestose and raffinose. Thus, the increased amount of LTA in cells cultured with raffinose cannot be explained by the levels of GroP polymer synthesis. One possible reason is the increase in LTA with short GroP chains during culture with raffinose compared with glucose and kestose.

LTA of Gram-positive bacteria, especially probiotic lactobacilli, is involved in host immune responses. LTA upregulates or downregulates cytokine production in a concentration-dependent manner [15,45,46,47]. D-Alanine residues of LTA have been reported as one of the key structures for the immune effects [14,48,49,50]. This suggests that the changes in structure and/or amount of LTA due to available carbohydrates affect the interaction between bacterial cells and the host. This is important when considering the further application of synbiotics. Synbiotics is defined simply as a mixture of probiotics and prebiotics [51]. Synergistic beneficial outcomes are not included in the definition but are warranted to promote the field of synbiotics. Further studies are required to clarify the reason for the differences in apparent molecular sizes of LTA and their impact on the probiotic properties of *L. gasseri* and *L. paragasseri* observed in the present study.

## Figures and Tables

**Figure 1 microorganisms-09-01590-f001:**
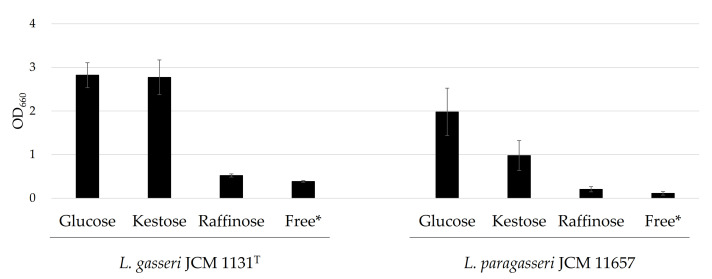
Growth characteristics of *L. gasseri* JCM 1131^T^ and *L. paragasseri* JCM 11657 cultured in the presence of glucose, kestose, or raffinose. Bars show averaged cell growth expressed as optical density at 660 nm (OD_660 nm_), and error bars indicate standard deviations (n = 3). ^*^ Sugar-free medium.

**Figure 2 microorganisms-09-01590-f002:**
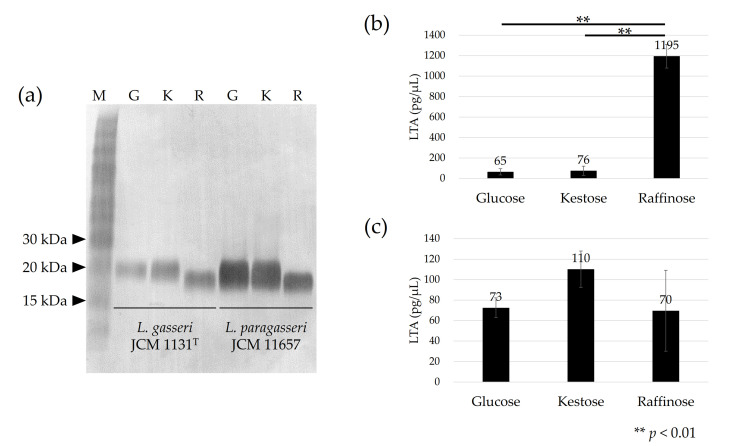
Immunoblot detection of lipoteichoic acid (LTA) in *L. gasseri* JCM 1131^T^ and *L. paragasseri* JCM 11657 cells (**a**), and amount of LTA in 1 µL of suspended (OD_660_
_nm_ = 1) *L. gasseri* cells (**b**), and *L. paragasseri* cells (**c**) cultured with glucose (denoted by G), kestose (denoted by K), or raffinose (denoted by R). Lane M in (**a**) is the size marker. Each lane in (**a**) is the μL of cell suspensions (OD_660 nm_ = 5) was used for each test, except OD_660 nm_ = 0.5 was used for *L. gasseri* cells cultured with raffinose because of the high LTA quantity. One-way ANOVA and Tukey’s post hoc test were applied to compare the amount of LTA among the cells cultured with different carbohydrates.

**Figure 3 microorganisms-09-01590-f003:**
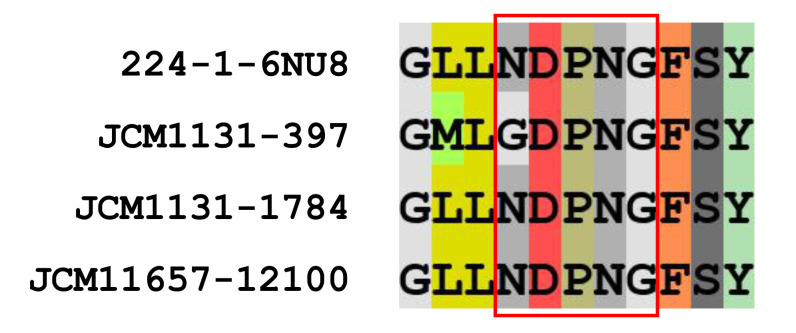
Sequences around the NDPNG motif (red boxed) of glycoside hydrolase (GH) 32 sucrose-6-phosphate hydrolase (S6PHs) found in *L. gasseri* JCM 1131^T^ and *L. paragasseri* JCM 11657. GH32 S6PH of *L. gasseri* strain 224-1, whose crystal structure has been determined (PDB ID, 6NU8), was included as a reference.

**Figure 4 microorganisms-09-01590-f004:**
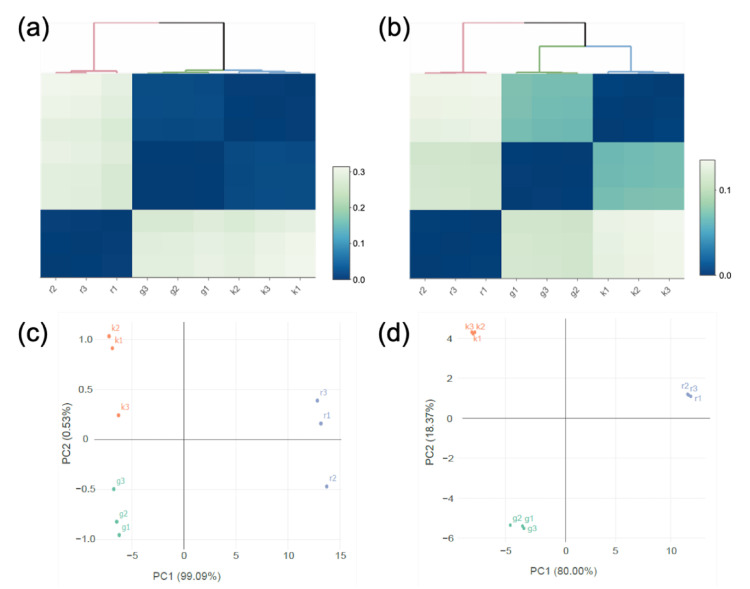
Hierarchical clustering (**a**) and (**b**) and Principal component analysis (PCA, **c** and **d**) based on transcriptome analysis of *L. gasseri* JCM 1131^T^ (**a**) and (**c**) and *L. paragasseri* JCM 11657 (**b** and **d**) in cultures with glucose (denoted by g), kestose (denoted by k), or raffinose (denoted by r). The transcriptome analysis was performed in triplicate using RNA isolated from three different cultures for each carbohydrate.

**Table 1 microorganisms-09-01590-t001:** Number of upregulated genes in the presence of different carbohydrates.

	Glc vs. Kes		Glc vs. Raf		Kes vs. Raf	
	Glc ↑ ^a^	Kes ↑	Glc ↑	Raf ↑	Kes ↑	Raf ↑
JCM 1131^T^	38	31	650	404	679	405
Sum	69		1054		1084	
JCM 11657	231	237	348	379	286	343
Sum	468		727		629	

^a^ Number of genes upregulated in the presence of glucose when compared with kestose.

**Table 2 microorganisms-09-01590-t002:** Number of upregulated genes in each Cluster of Orthologous Group (COG) class by the presence of different carbohydrates.

			C	D	E	F	G	H	I	J	K	L	M	N	O	P	Q	R	S	T	U	V	X
JCM 1131^T^		All CDSs ^a^	46	25	87	57	148	42	47	185	107	121	97	2	63	57	8	98	121	49	14	56	33
	Glc vs. Kes	Glc	2	0	2	1	10	0	1	3	2	0	2	0	0	0	0	3	2	2	0	0	0
		Kes	1	0	3	0	3	0	0	1	4	0	6	0	1	1	0	0	1	1	0	3	0
	Glc vs. Raf	Glc	11	8	33	24	33	17	22	83	34	48	33	0	18	18	3	34	52	18	4	21	7
		Raf	8	4	14	7	52	5	6	19	27	24	19	0	12	15	1	25	19	9	2	10	14
	Kes vs. Raf	Kes	16	8	38	21	45	13	17	85	29	52	37	1	21	20	2	40	49	19	7	21	8
		Raf	6	4	11	8	47	6	7	21	31	26	19	0	10	11	1	23	19	7	2	12	13
JCM 11657		All CDSs	45	27	95	60	176	39	45	190	106	111	97	3	64	58	8	105	126	52	17	57	17
	Glc vs. Kes	Glc	3	1	12	2	36	6	2	31	10	25	5	2	5	11	1	17	14	6	1	4	0
		Kes	3	1	23	17	12	7	6	6	13	18	9	0	9	12	2	12	15	6	3	5	1
	Glc vs. Raf	Glc	9	3	22	7	15	14	9	69	12	32	14	0	11	7	3	24	22	7	3	11	0
		Raf	15	3	8	13	60	1	6	13	39	15	27	0	8	13	0	25	16	10	5	12	4
	Kes vs. Raf	Kes	12	0	29	14	16	12	12	29	11	17	14	0	12	11	1	10	15	7	3	9	0
		Raf	18	2	7	7	92	1	4	19	32	11	16	2	4	11	0	16	11	8	3	9	2

^a^ Columns for all coding sequences (CDSs) are the number of genes assigned to each COG class using all CDSs in each strain. C: Energy production and conversion; D: Cell cycle control, cell division, chromosome partitioning; E: Amino acid transport and metabolism; F: Nucleotide transport and metabolism; G: Carbohydrate transport and metabolism; H: Coenzyme transport and metabolism; I: Lipid transport and metabolism; J: Translation, ribosomal structure and biogenesis; K: Transcription; L: Replication, recombination, and repair; M: Cell wall/membrane/envelope biogenesis; N: Cell motility; O: Posttranslational modification, protein turnover, chaperones; P: Inorganic ion transport and metabolism; Q: Secondary metabolites biosynthesis, transport, and catabolism; R: General function prediction only; S: Function unknown; T: Signal transduction mechanisms; U: Intracellular trafficking, secretion, and vesicular transport; V: Defense mechanisms; X: Mobilome, prophages, transposons.

**Table 3 microorganisms-09-01590-t003:** Comparison of gene expression levels in *L. gasseri* JCM 1131^T^ cells in the presence of different carbohydrates.

Gene ID	Product	Glucose (G) vs. Kestose (K)			Glucose (G) vs. Raffinose (R)			Kestose (K) vs. Raffinose (R)		
		log_2_ FC (G/K)	*p*	Significant Difference	log_2_ FC (G/R)	*p*	Significant Difference	log_2_ FC (K/R)	*p*	Significant Difference
1268	Levansucrase	−4.4	<0.01	K > G	−2.5	<0.01	R > G	1.9	<0.01	K > R
1269	levansucrase (GH68)	−4.7	<0.01	K > G	−2.3	<0.01	R > G	2.4	<0.01	K > R
396	PTS β-glucoside transporter subunit IIBCA	0.7	<0.01	- ^a^	−1.7	<0.01	R > G	−2.4	<0.01	R > K
397	sucrose-6-phosphate hydrolase (GH32)	0.3	0.01	-	−0.5	<0.01	-	−0.8	<0.01	-
1783	PTS β-glucoside transporter subunit IIBCA	−2.9	<0.01	K > G	−3.7	<0.01	R > G	−0.7	<0.01	-
1784	sucrose-6-phosphate hydrolase (GH32)	−1.1	<0.01	K > G	−2.2	<0.01	R > G	1.0	<0.01	K > R
0256	α-galactosidase (GH36)	−0.2	<0.01	-	0.1	0.03	-	0.3	<0.01	-
1820	D-alanyl transfer protein DltD	0.2	<0.01	-	0.6	<0.01	-	0.4	<0.01	-
1821	D-alanyl carrier protein DltC	0.2	0.01	-	0.7	<0.01	-	0.5	<0.01	-
1822	D-alanyl transfer protein DltB	0.2	0.01	-	1.1	<0.01	G > R	0.9	<0.01	-
1823	D-alanine-activating enzyme DltA	0.1	0.02	-	1.4	<0.01	G > R	1.2	<0.01	K > R
1824	D-Ala-teichoic acid biosynthesis protein DltX	0.1	0.24	-	1.5	<0.01	G > R	1.4	<0.01	K > R
1590	phosphoglycerol transferase (LtaS)	0.0	0.31	-	0.8	<0.01	-	0.7	<0.01	-
1591	membrane protein	0.2	0.18	-	0.9	<0.01	-	0.7	<0.01	-
1592	glycosyl transferase (GT4)	0.2	0.09	-	2.1	<0.01	G > R	1.9	<0.01	K > R
1593	glycosyl transferase (GT4)	0.2	0.05	-	2.6	<0.01	G > R	2.4	<0.01	K > R

- ^a^, no significant differences.

**Table 4 microorganisms-09-01590-t004:** Comparison of gene expression levels in *L. paragasseri* JCM 11657 cells in the presence of different carbohydrates.

Gene ID	Product	Glucose (G) vs. Kestose (K)			Glucose (G) vs. Raffinose (R)			Kestose (K) vs. Raffinose (R)		
		log_2_ FC (G/K)	*p*	Significant Difference	log_2_ FC (G/R)	*p*	Significant Difference	log_2_ FC (K/R)	*p*	Significant Difference
5790	Levansucrase	−4.2	<0.01	K > G	−2.7	<0.01	R > G	1.4	<0.01	K > R
5800	levansucrase (GH68)	−4.9	<0.01	K > G	−2.6	<0.01	R > G	2.2	<0.01	K > R
12100	sucrose-6-phosphate hydrolase (GH32)	−2.6	<0.01	K > G	−1.2	<0.01	R > G	1.5	<0.01	K > R
12110	PTS system, sucrose-specific IIABC component	−3.9	<0.01	K > G	−3.7	<0.01	R > G	0.2	0.01	-^a^
9630	sucrose phosphorylase (GH13)	−3.1	<0.01	K > G	−9.8	<0.01	R > G	−6.7	<0.01	R > K
9640	beta-glucoside-specific PTS system IIABC component	−1.8	<0.01	K > G	−8.6	<0.01	R > G	−6.8	<0.01	R > K
17400	α-galactosidase (GH36)	0.1	0.30	-	1.0	<0.01	G > R	0.9	<0.01	-
14610	oligo-1,6-glucosidase (GH13)	6.2	<0.01	G > K	1.2	<0.01	G > R	−5.0	<0.01	R > K
14620	sugar ABC transporter permease protein	6.4	<0.01	G > K	−0.1	0.33	-	−6.5	<0.01	R > K
14630	sugar ABC transporter permease protein	6.4	<0.01	G > K	−0.1	0.33	-	−6.5	<0.01	R > K
14640	sugar ABC transporter substrate-binding protein	6.6	<0.01	G > K	0.2	0.07	-	−6.4	<0.01	R > K
14650	glycerol-3-phosphate ABC transporter ATP-binding protein	6.8	<0.01	G > K	0.5	<0.01	-	−6.3	<0.01	R > K
14660	beta-phosphoglucomutase	7.1	<0.01	G > K	0.5	<0.01	-	−6.6	<0.01	R > K
14670	maltose phosphorylase (GH65)	6.8	<0.01	G > K	0.3	<0.01	-	−6.6	<0.01	R > K
14680	alpha-amylase (GH13)	3.8	<0.01	G > K	1.3	<0.01	G > R	−2.5	<0.01	R > K
14690	alpha-glucosidase (GH13)	3.6	<0.01	G > K	0.9	<0.01	-	−2.7	<0.01	R > K
10260	D-Ala-teichoic acid biosynthesis protein DltX	−0.1	0.29	-	2.1	<0.01	G > R	2.2	<0.01	K > R
10270	D-alanine-activating enzyme DltA	0.5	<0.01	-	2.0	<0.01	G > R	1.6	<0.01	K > R
10280	D-alanyl transfer protein DltB	1.2	<0.01	G > K	2.4	<0.01	G > R	1.2	<0.01	K > R
10290	D-alanyl carrier protein DltC	1.2	<0.01	G > K	2.1	<0.01	G > R	0.9	<0.01	-
10300	D-alanyl transfer protein DltD	1.4	<0.01	G > K	2.0	<0.01	G > R	0.6	<0.01	-
18430	phosphoglycerol transferase (LtaS)	0.3	<0.01	-	0.2	<0.01	-	−0.1	0.04	-
18440	membrane protein	0.6	<0.01	-	0.7	<0.01	-	0.1	0.19	-
18450	glycosyl transferase (GT4)	0.2	0.01	-	0.5	<0.01	-	0.2	<0.01	-
18460	glycosyl transferase (GT4)	−0.1	0.04	-	-0.0	0.57	-	0.1	0.14	-

- ^a^, no significant differences.

## Data Availability

The draft genome sequence of *L. paragasseri* JCM 11657 was deposited into the DDBJ/EMBL/GenBank under accession number BOPQ01000000. Raw RNAseq read sequences are available at DDBJ Sequence Read Archive (DRA) under the accession number DRA011689.

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
