# Peer review of "Oligosaccharide Metabolism and Lipoteichoic Acid Production in *Lactobacillus gasseri* and *Lactobacillus paragasseri"

_microorganisms, 2021, doi:10.3390/microorganisms9081590_

Round 1

Reviewer 1 Report

The authors studied the effect of different carbohydrates on the production  lipoteichoic acid (LTA) and characterized oligosaccharide metabolic systems of L. gasseri JCM 1131and L. paragasseri JCM 1165, employing transcriptomic analysis.   The methodology used is adequate for the objective proposed and the results were analyzed using appropriate statistical tools. The discussion is adequate and supported by updated bibliographies. The manuscript presents results that are important in the field of biotcs (Probiotics, prebiotics and synbiotics). I suggest the publication of the manuscript in Microorganisms with minor revisions.

Some suggestions:

- In the abstract section, page 1, lines 27-28 authors should add the numerical value of the time that production increased LTA

- The abstract should end by explaining the importance of research conducted.

- In the introduction section page 2, lines 81-84. Authors should rewrite this sentence, focusing on the purpose of their research.

- The phrase: “….Raffinose, a common prebiotic, was used for a reference. RNA sequencing was conducted to characterize carbohydrate metabolic systems and LTA production systems in these strains”,  should be omitted from this section and should be added in the materials and methods section.

-The authors did not consider conducting the production and Immunoblot detection of LTA on cell suspensions from free-sugar medium?

Reviewer 2 Report

This is a very good paper based on up-to-date methodology, with sound results and nicely presented. The reviewer has, however, some suggestions:

- The authors may consider dividing the content into two parts: one on oligosaccharides metabolism and second on LTA production by the two tested collection strains. Although both activities were followed with the same transcriptomic analysis under influence of  the same carbohydrates, their meaning is not the same, as oligosaccharides produced by the strains may be regarded as "internal" prebiotics, while LTA not. Morever, as the authors mentioned in discussion, little is known about Lactobacillus LTA in comparison to this of S.aureus. Term "immunomodulation" used in this context has a very broad meaning rather not related to prebiotics and synbiotics.             
